# Differences between Elite and Professional Male Handball Players in Kinematic Parameters of Single Fake Movement

**DOI:** 10.3390/jfmk8020047

**Published:** 2023-04-20

**Authors:** Ante Burger, Dario Vrdoljak, Nikola Foretić, Miodrag Spasić, Vladimir Pavlinović

**Affiliations:** 1University Department of Health Studies, University of Split, 21000 Split, Croatia; 2Faculty of Kinesiology, University of Split, 21000 Split, Croatia; darvrd@kifst.hr (D.V.);

**Keywords:** team handball, feint movement, biomechanics

## Abstract

Feint movement is an important factor for offensive players to outplay their guard, and score. So far, there is no evidence of feint biomechanical analysis on a sample of elite players in handball or other team sports. Therefore, this study aimed to investigate kinematic parameters of single side fake movement between elite and professional level handball players. The sample of participants consisted of 10 handball players divided into two subsamples: elite handball players (100.00 ± 8.00 kg; 196.00 ± 4.64 cm) and professional handball players (91.20 ± 3.42 kg; 192.4 ± 7.30 cm). The kinematic analysis was conducted using a GAIT—LaBACS software system. Variables consisted of two phases (fake phase and actual phase) of feint single change of direction. Both phases included seven kinematic parameters that were observed. Statistical analysis included descriptive statistic parameters. The differences between elite and professional handball players were analyzed by multivariate and univariate variance analysis. Results showed significant differences between elite and professional players (λ = 0.44, *p* = 0.00), in fake phase (i.e., 1. Phase). The results also indicate that in there is no statistically significant difference between both groups (λ = 0.64, *p* = 0.22). Two variables had significant differences between elite and professional players (i.e., step length of the stride leg (*p* = 0.02) and moving the leg opposite the throwing arm in space (*p* = 0.00)). To conclude, the article examines specific movement patterns of single side fake movement in elite players and the confirmed importance of efficient skill execution in top level handball. On the contrary, less skilled players use more space for the same technical element.

## 1. Introduction

Handball is a technical-tactically, physically and psychologically complex sport that requires full engagement from players [1,2]. Factors that influence the performance of handball players are numerous, because of its complexity and structure. The most important of the factors is an efficient execution of specific tasks, which is an aggregation of running, jumping and direction changes with sport-specific passing, catching, throwing and physical contact [3,4]. Additionally, handball coaches assess a player’s offensive quality by his ability to outplay his guard. This is carried out with shoots or fake movements/feints. While shoots are used to finish the attack, feints are the most important elements for gaining an advantage over a defender and making cracks in defense [5]. In modern handball, defensive zone formations are set very deep with strong and explosive defenders [6]. Shooting on goal has become very difficult to achieve just through the individual actions of the attacker. Hence, shooting must precede an advantage that another player gained. Gained advantage may end with a direct shoot of player who performs a feint. Mostly, due to the defensive corrective actions (such as overtaking and helping), an attacker passes the ball and gains advantage during the fake movement, thus transferring the ball to a player with a better position for shooting into the goal. From this reason, it is expected that during the handball match, there are significantly more fake movements than shots on goal.

There are many fake movements that attackers use to trick defenders, but most complex are feints with the ball. These movements are commonly used by handball players, but are lacking any scientific approach. The feint single change of direction belongs to the attacking technique with the ball. Rogulj et al. [7] define feints as a technical element by which a player in attack achieves a temporal or spatial advantage over a defensive player who guards him. This motion allows him to carry out undisturbed realization or ensure continued technical and tactical action in more favorable conditions than the opponent [8]. Feints should be carried out when the defensive players least expect it. Additionally, according to Vuleta et al. [9], center back players’ high level of performance is defined by good feint execution.

A review of the aforementioned aspects leads to the assumption that the execution technique of those skills plays an important role in success determination. Moreover, apart from anthropology, motor and functional skills, individual technique plays an important role in differentiation of elite and non-elite players [10,11,12]. Lanka et al. [13] describe the technique as executive motor-neuromuscular activity and motor realization of imaginary movement in accordance with basic biomechanical principles. It is important to note that the ideal technique may not be the best for every player as they differ from each other in motor and morphological features. Furthermore, some studies showed that difference between non-elite and elite handball players can be found in the use of techniques, especially in throwing motions [14].

So far, the feint technique has been analyzed only as a sport-specific test [15] or in teaching programs for beginners [16]. The importance of feint execution in other team sports was detected in several previous studies, one of which is basketball, where players use one-on-one feints with dribbles and body movements in offensive situations against opponents [17]. Apart from that, dribbling with feints is also used by basketball players [18]. Both studies gave an informational model for performance enhancement in game situations. Moreover, Güldenpenning et al. [19] showed how elite and novice athletes in volleyball differ in feint recognition. Authors concluded that elite athletes were able to predict feint attack at an earlier stage, and therefore showed the importance of good feint execution at high level of sport.

However, there is no evidence of feint biomechanical analysis on elite players sample in handball or other team sports. Following all aforementioned, this study aimed to investigate kinematic parameters of single side fake movement between elite and professional level handball players. More specifically, step lengths, angles between joints and movement speed will be measured by kinematic analysis. 

## 2. Materials and Methods

### 2.1. Participants

The sample of participants consisted of 10 handball players divided into two subsamples: elite (representative) and quality (first league) players. The selection criteria for elite players was participation in at least five major top-level handball competitions (European Championship, World Championship or Olympic Games). The selection criteria for quality players was participation in at least three seasons in the First Division National Championship. For the sake of time efficiency, all players were recruited from wider surroundings of the city of Split during the offseason period. Elite handball players (representative sample) were represented by five players who played for the Croatian national handball team and won medals in the Olympic Games, World and European handball championships (body mass: 100.00 ± 8.00 kg, body height: 196.00 ± 4.64 cm, four back players and one wing player). Professional handball players (first league sample) were represented by five players who have a minimum of three seasons in their handball career in the First Croatian Handball League (body mass: 91.20 ± 3.42 kg, body height: 192.4 ± 7.30 cm, four back players and one wing player). Informed consent was obtained from all subjects involved in the study. Experimental procedures were completed following the declaration of Helsinki. All athletes participated in handball training on a daily basis, which had a significantly higher risk than the testing procedure conducted. All of them were aware of minimal risk identified and voluntarily participated in the study. Hence, there was no need for ethical board approval.

### 2.2. Design and Procedures

The kinematic analysis was conducted using a GAIT—LaBACS software system (ver. 1.0, Split, Croatia) designed at the Faculty of Electrical and Mechanical Engineering and Naval Architecture in Split [20]. Variables consisted of two groups that correspond to two phases of feint single change of direction. This element is divided into two parts, the fake phase and the actual (executive) phase. The fake phase had seven kinematic parameters: step length of the stride leg (SLS), the speed of the false part of the phase (TFP), angle of the trunk in relation to the ground (ATG), scrolling the center of gravity of the body-(SCG), duration of the false part of the phase (DFP), moving the leg opposite the throwing arm in space (MOT) and the position of the foot that is opposite of the throwing arm at the end of the actual phase (FOA). The actual (executive) phase had seven kinematic parameters: total length of all steps (TLS), length of step 1 (LS1), length of step 2 (LS2), speed of the first step (S1S), speed of the second step (S2S), direction of step 2 (DS2) and duration of the actual (executive) phase (DAP).

For the purposes of kinematic analysis, twelve symmetric reference points were defined: The reference point is placed at the lower end of the outer ankle (malleoli); the reference point is placed in the middle of the knee joint from the front at the chip position (patella); the reference point is placed at the top of the pelvic bone at the point of the (cresta iliaca); the reference point is set on the (acromion); the reference point is placed at the end of the ulnar bone (olecranon); the reference point is placed at the ends of the ulnar and radial bones from the anterior side (Figure 1). The recording was performed in the sport hall, with a previous calibration of the measuring system, while the layout of the cameras was such that all twelve reference points on the body of the subjects were visible. The cameras used were BASLER-402-FC (Dortmund, Germany), reproducing 100 pictures per second, and PANASONIC VW-D5100 reproducing 50 pictures per second. First of all, the measuring space had to calibrated. Calibration was performed by placing an object of known dimensions in the measurement space. After that, the XY values of the reference points on that object were read (from the camera images). By entering the read values from the images, as well as the spatial coordinates of reference points, it is possible to use the DLT (direct linear transformation) method to form a matrix used to calculate spatial coordinates for every other point in the measurement space. 3D calibration of the space was performed by placing eight static markers, where four were positioned in the vertical direction on a metal rod with a defined height, and four positioned on the ground, arranged at the bottom of the metal rod in a proportional square spacing. Video sequences were reviewed, and reference point positions were read and recorded from each camera. Using the DLT method, the spatial position of the reference points at each time point of the element design was calculated. For each analyzed element, the recording was repeated three times. The analysis of the quality of the recordings was carried out immediately after the testing, so that any errors in the video could be corrected or repeated with a new test.

### 2.3. Statistical Analysis

Statistical analysis included calculation of descriptive statistic parameters while normality was tested using the Kolmogorov–Smirnov test procedure. The differences in kinematic parameters between elite and professional handball players were analyzed by multivariate (used to determine differences between all parameters of the given fake movement phase) and univariate variance analysis (used to determine differences between each parameter of the given fake movement phase). The level of significance was set at p < 0.05. The software Statistica ver.13.0 (Dell Inc., Austin, TX, USA) was used for all analysis.

## 3. Results

The Kolmogorov–Smirnov test exhibited normal distribution of the variables (*p* > 0.20). The calculation of descriptive statistic parameters included means, standard deviations and minimum and maximum for both phases and groups.

Table 1 shows statistically significant differences between elite and professional handball players (λ = 0.44, *p* = 0.00), in fake phase (i.e., 1. Phase). The results also indicate that in the second statistically significant difference between groups, do not exist (λ = 0.64, *p* = 0.22).

Univariate analysis of variance (ANOVA) is presented in Table 2. From the given results it is visible that two variables had significant difference between elite and professional players (i.e., SLS (*p* = 0.02) and MOT (*p* = 0.00)). Elite players execute smaller steps (SLS) during the first phase of the fake movement (106.44 cm) than professional players (133.55 cm). MOT results implicate that elite players exhibit a lesser angle (17.70°) than professional players (31.42°).

Differences between groups in the second phase of the fake movement are presented in Table 3. In the variables of the second phase, there are no statistically significant differences recorded. However, some differences can be noticed. Specifically, elite players have more speed (4.44 m/s) in this part of the second phase than professionals (4.11 m/s). Additionally, in DS2, elite players have smaller angle of step (13.43°) than professional players (17.64°).

## 4. Discussion

This study aimed to investigate kinematic parameters of single side fake movement between elite and professional handball players. The obtained results indicate several important findings: (1) Players differ in the first phase of the feint; (2) elite players have a smaller first step and move more sideways in the first phase; and (3) differences in the second phase are connected to first phase performance but are not statistically significant.

The literature review refers to significant differences in movement pattern efficiency dependable on players’ skill levels. This was noticed between skilled and novice players in volleyball, soccer, basketball and tennis [17,21,22,23,24,25]. For example, Fujii, Yamada and Oda [21] demonstrated that skilled basketball players had smaller decrement in maximal sprint to maximal dribbling speed. Their results indicate greater compensation in dribbling through body segment movements. Moreover, basketball players also showed differences in ball handling [22]. It was shown that skilled players have longer and more consistent ball contact and, therefore, better spatial control of the ball. According to Loffing and Hagemann (2014), unlike the novice, skilled tennis players consider the reliability of different information sources by weighting the available contextual and kinematic cues differently in the course of an opponent’s unfolding action [25]. Furthermore, studies conducted in soccer show effective upper-body movement in skilled players to be a key factor in creating better initial conditions for a more explosive muscle contraction during kicking [24]. Obviously, it could be expected that, in sports, a higher level of sport competition demands more efficient technique and movement pattern performance. Efficient movement saves time, increases speed and power and therefore represents advantage to more efficient athletes [26]. Movement pattern efficiency and its dependence on skill level in handball was investigated in several studies. Those studies analyzed kinematic parameters and were mainly focused on overarm throwing and jump-shot performance [27,28,29,30]. In some of these studies, a lack of differences was observed between novice and expert players [27]. The authors showed that changing the goal of the task similarly effected velocity of the ball and movement of body segments in both groups. Therefore, authors concluded that, in this particular case, training experience is not related to overarm throwing performance. However, in jump-shots, technique differences were found. Elite players perform jump-shots differently. They had increased trunk flexion and rotational angular velocity, which results in an increase in ball release speed [28]. Making a large step (like professional players in this study) lower than his center of mass causes a wider angle of equilibrium. A wider angle of equilibrium implies better stability, which can be counterproductive in this situation after which an explosive countermovement is required. Furthermore, by taking such a large step, one put his lower extremities in a disadvantageous position for the next quick reaction, considering the angles in the joints and torques that the muscles can produce at these angles. Not to mention that a shorter step also takes less time to perform, and it is known that time is a significant factor when performing a successful feint.

Our results correspond to previously mentioned studies in terms of differences between elite and professional players, as some of kinematic parameters differ between the groups. Still, it was evident only in the first phase of the fake movement. Results show that elite players performed specific movement pattern with the shorter sidestep for 27.11 cm than professional players.

Additionally, elite players conduct fake movements more to the side, unlike professional players who perform them more diagonally. This can be noticed from the MOT variable in which professional players during the fake movement have a larger angle of 13.72° between the feet and frontal plane. The diagonal position of the attackers’ feet puts the offensive player in less favorable position towards the defender. This way the attacker is closer to defender and has less space for executive phase of fake movement that should be performed faster and more explosively. In addition, by thrusting too far forward, the attacker leaves the defender more sideways, and in order to pass him on the other side in the next moment (which is the next phase of the feint), it is necessary to go “back” and then go around the defender. Again, we see that time is important for performing a successful feint. The defender’s job is to stop the attacker with tackles. In those tackles, he always tries to be as close as possible to the attacker [31]. Hence, we can conclude that elite players’ movements in the first phase is more efficient in real game situations. Moving the leg more sideways allows the player to shift his center of gravity without losing balance while also maintaining the strong support that is necessary for the agile and explosive second phase of feint [32]. Shifting the center of gravity is important for the reaction of the defender because an experienced defender should react only to a significant change in the position of the center of gravity of attacker and not to feinting (only) with the extremities. This is supported by the fact that the best representation of the body is its center of gravity If one wants to describe trajectory of body’s movement in general, he or she can use that specific point which would substitute the whole body [33] A bigger shift sideways forces the defender to react with a more resolute tackle. Tackles of this kind are more advantageous for defenders, since they have to move more and faster. Rapid dropping out of defending position is critical for defensive stance, body balance and consequently losing the duel with the attacker. Although statistical analysis did not show significant differences between the two groups, it is evident that elite players use less space in the second phase of the feint. Specifically, total length of all steps, length of the first and second step separately, are shorter in elite than in professional players, respectively. Efficiency of the second phase is directly influenced by first phase and, most probably, by first phase execution. As stated earlier, the elite players perform it in a smaller space than professional players. Hence, they need less space and distance (and consequently time) to finish it in the executive phase of the feint. Overall, it can be stated that elite players perform single side fake movement more efficiently than professional players.

This study had several considerable limitations. Although players in two groups played the same playing position, we noticed a significant difference in anthropometrical indices that could influence skill execution. In several studies, positive correlations were noticed between the body mass and body height and throwing kinematic parameters. Body mass influenced speed variables while body height influenced the height of the body’s center of gravity [10,28]. Similar influences are expected in our sample and contribute to the possible distribution of data homogeneity.

Additionally, lack of some more precise anthropometric measurements could be influential in the assessment of feint movement. Furthermore, in this study we have researched only kinematic parameters of single side fake movement. Future studies should include other factors that are important in handball players’ feinting performance. Two aspects should be specially researched: (1) cognitive abilities of players, such as perceptual and decision-making factors, and (2) physical abilities of players, such as timing, muscle quality, strength, power and/or reactive agility factors. It would be important to know which of these to is more important for successful handball feint performance.

## 5. Conclusions

Research of kinematic parameters of single side fake movement for feint single change of direction has not been carried out so far, not just in handball but in team sport games generally. Hence, it was impossible to compare the obtained results and methodology with literature review. Results show specific movement pattern of single side fake movement in elite players and confirmed importance of efficient skill execution in top level handball. On the contrary, less skilled players use more space for the same technical element. Results of the study could direct handball coaches in better understanding single side fake movement performance. During teaching this important element they should focus on: (1) optimal step length; players’ needs to adapt first step length to a defender’s position and his anthropometry, (2) maintaining a straight body position while shifting center of gravity; during teaching the single side fake movement strait body position is crucial for controlling body balance and good perception of the game and (3) controlling the distance from defender; conducting single side fake movement to close to defender is inefficient, so the attacker should be learned to avoid physical contact and perform it on 5 to 10 cm distance of the defender’s arms reach.

## Figures and Tables

**Figure 1 jfmk-08-00047-f001:**
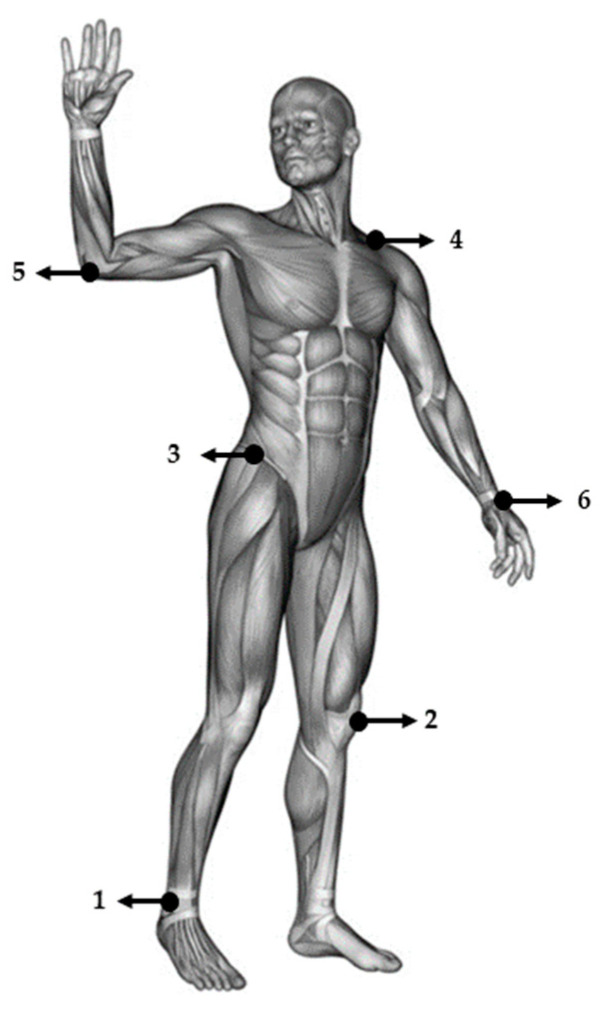
Reference points for kinematic analysis, defined as: (1) malleoli; (2) patella; (3) crista iliaca; (4) acromion; (5) olecranon; (6) end of ulnar and radial bones.

**Table 1 jfmk-08-00047-t001:** Differences between elite and professional handball players in parameters of single fake movement calculated by MANOVA.

1. Phase–fake phase			
Test	Wilks λ	F	*p*
λ	0.44	4.64	0.00 *
2. Phase–actual (executive phase)			
Test	Wilks λ	F	*p*
λ	0.64	1.5	0.22

λ—Wilk’s lambda, F—F test, *p*—Coefficient of significant difference, *—*p* < 0.05.

**Table 2 jfmk-08-00047-t002:** Differences between elite and professional handball players in the first phase of fake movement calculated by ANOVA.

	Mean Elite Players	Mean Professional Players	F	*p*
SLS	106.44 (cm)	133.55 (cm)	5.81	0.02 *
TFP	5.16 (m/s)	5.76 (m/s)	1.13	0.30
ATG	74.35 (°)	75.41 (°)	0.18	0.67
SCG	78.12 (cm)	93.17 (cm)	3.6	0.07
DFP	0.21 (ms)	0.25 (ms)	2.17	0.15
MOT	17.70 (°)	31.42 (°)	11.87	0.00 *
FOA	95.00 (°)	89.00 (°)	3.33	0.07

SLS—step length of the stride leg, TFP—the speed of the false part of the phase, ATG—angle of the trunk in relation to the ground, SCG—scrolling the center of gravity of the body, DFP—duration of the false part of the phase, MOT—moving the leg opposite the throwing arm in space, FOA—the position of the foot that is opposite of the throwing arm at the end of the fake phase, F—F test, *p*—Coefficient of significant difference, *—*p* < 0.05.

**Table 3 jfmk-08-00047-t003:** Differences between elite and professional handball players in the second phase of the fake movement calculated by ANOVA.

Variables	Mean Elite Players	Mean Professional Players	F	*p*
TLS	328.44 (cm)	342.78 (cm)	0.54	0.46
LS1	150.95 (cm)	162.08 (cm)	0.76	0.38
LS2	177.56 (cm)	180.51 (cm)	0	0.95
S1S	5.41 (m/s)	5.41 (m/s)	0.01	0.91
S2S	4.44 (m/s)	4.11 (m/s)	1.96	0.17
DS2	13.43 (°)	17.64 (°)	2.48	0.13
DAP	0.77 (ms)	0.80 (ms)	0.24	0.62

TLS—total length of all steps, LS1—length of step 1, LS2—length of step 2, S1S—speed of the first step, S2S—speed of the second step, DS2—direction of step 2, DAP—duration of the actual (executive) phase, F—F test, *p*—Coefficient of significant difference.

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
