# Peer review of "Differences between Elite and Professional Male Handball Players in Kinematic Parameters of Single Fake Movement"

_jfmk, 2023, doi:10.3390/jfmk8020047_

Round 1

Reviewer 1 Report

The methodological aspect of the study, specifically the sample, is valued very favorably.

Being able to analyze athletes with victories in Olympic games or world championships is very complicated. Although the sample may seem small, from my point of view, it is sufficiently representative of the population at this level.

With regard to the rationale in the introduction and even in the discussion, the importance of the feint action in the real game is missing.

It is not one more technical action, but from the field of coaches, it is a determining aspect in the performance of team sports players.

Please contextualize this fact in the discussion.

Fasold, F., Meyer, J., & Klatt, S. (2023). Effects of throwing feints on attack strategy in male elite handball: a post-hoc video analysis. International Journal of Performance Analysis in Sport, 1-12.

Büchel, D., Gokeler, A., Heuvelmans, P., & Baumeister, J. (2022). Increased Cognitive Demands Affect Agility Performance in Female Athletes-Implications for Testing and Training of Agility in Team Ball Sports. Perceptual and Motor Skills129(4), 1074-1088.

Author Response

Dear reviewer,

Thank you for your comments. We have revised and corrected the manuscript accordingly. Therefore, the answers are attached.

Reviewer 2 Report

Interesting idea of the article, my recommendations are the following:

In abstract

- I recommend that you mention how the kinematic analysis was carried out.

- clearly mention the two phases of the slit.

Line 14 - the false phase has 7 kinematic parameters and the other??? I recommend clarification.

I recommend rewriting the conclusions, they are not clear.

Line 240 Conclusions recommend that:

The limitations should be moved to the end of the Discussions section.

It is not advisable to repeat the aimed and novelty again.

Conclusions should be rewritten focused on results.

The article is well organized and presented with the exception of the abstract and conclusions.

Author Response

(The authors gave the same response as above.)

Reviewer 3 Report

Thank you for the opportunity to review this article. The paper addresses a novel under-researched area, which has the potential to provide useful recommendations for coaches. However, there are some questions that need to be addressed to the manuscript.

Specific comments are provided below:

TITLE

Include the gender sample in the title “elite and professional male handball players”. 

INTRODUCTION

-       The text jumps between different topics (e.g., factors that influence performance, feint execution, technique, previous research in other team sports) without a clear structure. It might be helpful to reorganize the text to provide a more logical and coherent flow of ideas. For example, the text could start with a brief overview of handball and factors that influence performance, then focus specifically on feint execution and its importance in handball, and finally discuss previous research and the research question of the current study.

-       While the text states that the study aims to investigate kinematic parameters of single side fake movement between elite and professional level handball players, it would be helpful to state the research question more explicitly. For example, what specific kinematic parameters are being investigated? How do these parameters differ between elite and professional players?

-       Although the text touches upon the basics of handball and feint execution, it lacks sufficient context to help readers comprehend the importance of the research question. It would be helpful to include more information about the existing research on feint execution in handball and other team sports, in order to provide readers with a clearer understanding of the significance of the topic.

MATERIALS AND METHODS

-       Clarify the selection criteria for the participants. Why were these particular individuals selected to participate in the study? (line 71-80)

-       Provide more detail on the recruitment process. How were the participants recruited? (line 71-80)

-       Add how ethical considerations were addressed in the study, such as obtaining informed consent from the participants and the ethics committee code (line 71-80)

-       Explain the significance of conducting a kinematic analysis and why these specific kinematic parameters were chosen for the study. (line 82-120)

-       Provide more detail on the specific equipment and cameras used for the kinematic analysis. (line 82-120)

-       Include a diagram or illustration of the reference points and measuring system used for the analysis. (line 82-120)

-       Provide more information about the specific tests used for the multivariate and univariate variance analysis. For example, you could mention whether you used ANOVA or MANOVA and how you set up the models. (line 121-127)

CONCLUSIONS

-       Please, include the limitation, future studies and practical implications in the discussion section. (line 248-260)

-       When discussing the limitations of the study, try to be more precise in identifying the specific limitations and their potential impact on the results. For example, instead of simply noting the differences in body height and body mass between the two groups, discuss how these differences might have affected the kinematic parameters and the interpretation of the results. (line 248-260)

-       Provide more specific recommendations for future research. Rather than just listing a number of factors that could be investigated, suggest specific hypotheses and research questions that could be addressed in future studies. (line 248-260)

-       When discussing the practical implications of the study, be more specific about how coaches can use the results to improve their training. Provide concrete examples of how coaches can focus on optimal step length, maintaining straight body position, and controlling the distance from defenders during training. (line 248-260)

Author Response

(The authors gave the same response as above.)
